# Atom-level interaction design between amines and support for achieving efficient and stable CO$_2$ capture

Xin Sun [1], Xuehua Shen [1,2] ✉, Hao Wang [3] ✉, Feng Yan [1,2], Jiali Hua[1], Guanghuan Li[1] & Zuotai Zhang [1,2,3] ✉

Amine-functionalized adsorbents offer substantial potential for CO$_2$ capture owing to their selectivity and diverse application scenarios. However, their effectiveness is hindered by low efficiency and unstable cyclic performance. Here we introduce an amine-support system designed to achieve efficient and stable CO$_2$ capture. Through atom-level design, each polyethyleneimine (PEI) molecule is precisely impregnated into the cage-like pore of MIL−101(Cr), forming stable composites via strong coordination with unsaturated Cr acid sites within the crystal lattice. The resulting adsorbent demonstrates a low regeneration energy (39.6 kJ/mol$_{CO2}$), excellent cyclic stability (0.18% decay per cycle under dry CO$_2$ regeneration), high CO$_2$ adsorption capacity (4.0 mmol/g), and rapid adsorption kinetics (15 min for saturation at 30 °C). These properties stem from the unique electron-level interaction between the amine and the support, effectively preventing carbamate products' dehydration. This work presents a feasible and promising cost-effective and sustainable CO$_2$ capture strategy.

Increasing CO$_2$ emissions have led to a cascade of environmental catastrophes. In response to this global challenge, over 100 nations have committed to achieving carbon neutrality by 2050 and limiting the average global temperature rise 1.5−2 °C above preindustrial levels[1,2]. Carbon capture, utilization, and storage (CCUS) emerge as the most promising and effective short-term solution for reducing CO$_2$ levels[3]. According to projections by the International Energy Agency, CCUS is expected to capture 27 billion tons of CO$_2$ by 2050 cumulatively[4]. Presently, the prevalent method for CO$_2$ capture from industrial flue gas involves using aqueous amine solutions worldwide[5]. However, this technology is prone to oxidation, degradation, and secondary pollution and entails substantial energy consumption during amine solution regeneration[6]. Therefore, exploring viable alternatives is imperative. Among these, porous solid adsorbents emerge as leading contenders for next-generation carbon capture technology due to their non-corrosive nature and lower energy requirements for regeneration[7,8].

Porous solid adsorbents, including amine-functionalized adsorbents[9,10], zeolites[11], and metal-organic frameworks[12], have been extensively studied for CO$_2$ capture. Amine-functionalized adsorbents, in particular, are highly selective and resistant to water vapor, rendering them suitable for diverse application scenarios involving humid and low-concentration CO$_2$ environments[2,13]. Typically, these adsorbents are prepared by impregnating polymeric amines such as polyethyleneimine (PEI) into porous materials[14,15], grafting aminosilanes[16] onto support surfaces, or via in situ polymerization[17] of amine monomers within supports. High loading of amines into large-porosity materials enables high CO$_2$ uptakes (≥3 mmol/g)[7,18]. Previous studies have predominantly focused on enlarging the support mesopores (2−50 nm) to enhance amine loading[19]. However, aggregation frequently occurs due to poor amine dispersion within the support. Large pore spaces allow multiple polymeric amine molecules to impregnate a single pore, thereby greatly increasing CO$_2$ diffusion resistance and

[1]School of Environmental Science and Engineering, Southern University of Science and Technology, Shenzhen 518055, China. [2]Key Laboratory of Municipal Solid Waste Recycling Technology and Management of Shenzhen City, Shenzhen 518055, China. [3]Hoffmann Institute of Advanced Materials, Shenzhen Polytechnic University, Shenzhen, Guangdong 518055, China. ✉e-mail: shenxh@sustech.edu.cn; wanghao@szpu.edu.cn; zhangzt@sustech.edu.cn

resulting in slow $CO_2$ adsorption rates[20]. These adsorbents often necessitate higher $CO_2$ adsorb/desorb temperatures, leading to increased energy consumption and capture costs[21]. Another challenge is the rapid deactivation of amines during regeneration in a $CO_2$-rich atmosphere, a critical step for separating and recycling pure $CO_2$[22]. Most amine-functionalized adsorbents lose their $CO_2$ capture ability after a few cycles due to dehydration reactions between the amines and $CO_2$-forming urea[23]. Addressing amine deactivation is thus crucial when considering associated costs and scalability challenges[24]. Therefore, the development of amine-functionalized adsorbents that minimize $CO_2$ diffusion restrictions during the adsorption process and exhibit strong resistance to deactivation in a $CO_2$-rich atmosphere is paramount.

To enhance amine dispersion within supports and mitigate aggregation, employing smaller micropores for amine impregnation can be an effective strategy[25]. However, previous studies have highlighted a substantial challenge in impregnating amines into micropores due to the close size match between amine molecules and micropores, resulting in considerable hindrance during the impregnation process[7,26,27]. Inspirational insights indicate that designing an atom-level interaction amine-support system with two key characteristics holds promising potential. First, the dispersion-focused characteristic involves the precise impregnation of individual amine molecules into specific micropores of the support material, possessing unique active sites, thereby overcoming impregnation resistance. Second, the stability-focused characteristic relates to the active sites enabling interactions with amines, forming a stable structure capable of resisting deactivation caused by urea formation[10,13]. Most supports are unsuitable for this amine-support system because their primary function is enhancing the accessibility of amines. Metal-organic frameworks offer exceptional control over pore size and surface chemistries[12]. Specifically, MIL−101(Cr) stands out due to its high porosity, customized pore structure, suitable pore dimensions, and the interconnected cages within MIL−101(Cr) exhibiting a high density of Cr acidic sites[28]. This renders it a promising support for the amine-support system. While previous studies have demonstrated the $CO_2$ capacity of MIL−101(Cr) as a porous support for amine impregnation[14,29], they did not delve into the design of an amine-support system with atomic-level precision to address hindered diffusion and amine deactivation.

In this study, we present an innovative approach involving the impregnation of amines into the crystal internalization (IACI) of MIL−101(Cr) to achieve uniform dispersions and high stability. To assess its potential for CCUS, we conducted measurements of $CO_2$ adsorption performance, regeneration energy, and cyclic stability under varying temperatures, pressures, and regeneration atmospheres. Additionally, we employed time-of-flight secondary ion mass spectrometry and density functional theory (DFT) to examine the structure and deactivation resistance of IACI. Our investigation revealed the electronic-level origin of IACI urea inhibition under a dry $CO_2$ regeneration atmosphere. Our findings demonstrate that IACI exhibits negligible diffusion resistance, with saturation adsorption achieved within a mere 15 min at 30 °C. Furthermore, it displays exceptional stability, with decay rates of 0.11% and 0.18% per cycle in Ar and pure $CO_2$ atmospheres, respectively, and a high adsorption capacity of 4.0 mmol/g at 1 bar and 278 K.

## Results

### Synthesis and characterization of IACI

The pristine porous support of MIL−101(Cr) was synthesized according to previous reports (Supplementary Fig. 1). It consists of cages with uniform 1.6-nm intracrystalline micropores[28] (Fig. 1a) and a high density of Lewis acid sites (Supplementary Fig. 2), rendering it an ideal candidate for the proposed amine-support system. To integrate cooperation between the amine and support, we employed various molecular-weight PEIs (Supplementary Note 1) to achieve an optimal match between the dimensions of the pores and amines (Supplementary Fig. 3). Among these amines, PEI-1200, with planar dimensions of 1.3 nm, was chosen as it perfectly fits the requirement of precisely accommodating a single PEI-1200 molecule within each pore to pass through the hexagonal windows in MIL−101(Cr) (Fig. 1a). To validate our hypothesis regarding the role of acidic Cr sites in MIL−101(Cr) cages as a driving force for the amine-support system within micropores, we synthesized PEI-functionalized MIL−101(Cr) using PEI-1200 and MIL−101(Cr) via moderate impregnation (Supplementary Fig. 4). As depicted in Fig. 1b and c, porosity analysis assessed changes in Brunauer−Emmett−Teller specific surface areas ($S_{BET}$) and pore volumes at various impregnation intervals. Initially, $S_{BET}$ decreased from 3468 to 1180 m²/g, with a reduction in pore volume from 1.75 to 0.63 cm³/g within the first 5 min. This indicated rapid loading of PEI-1200 into the MIL−101(Cr) support, occupying a

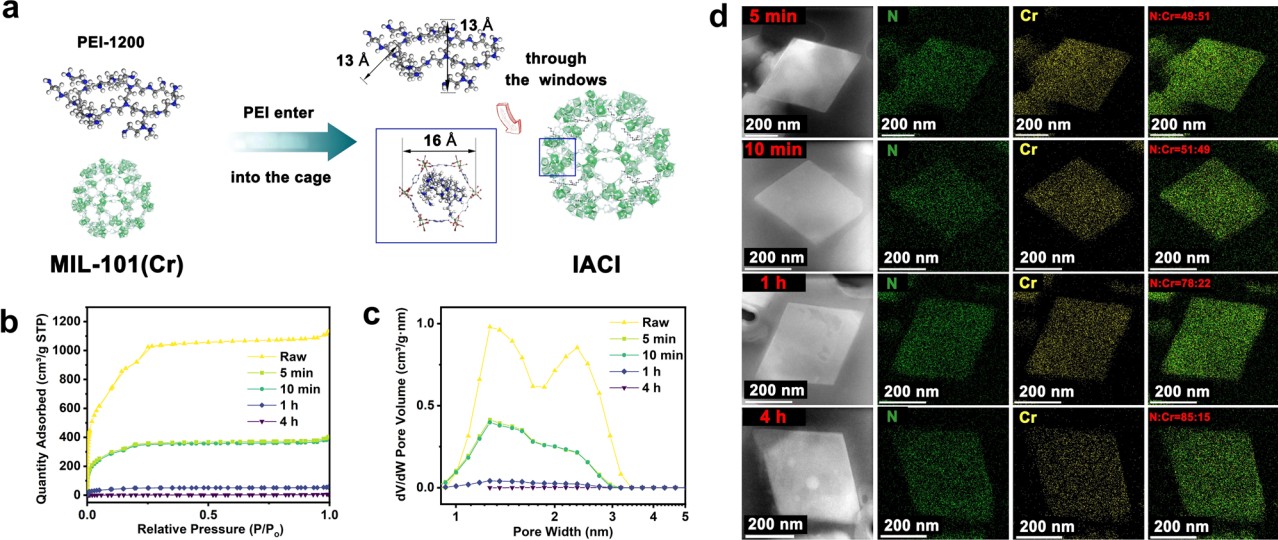

Fig. 1 | Characterization of impregnating amines into crystal internalization (IACI) in PEI-functionalized MIL−101(Cr). a Schematic of IACI. b $N_2$ adsorption/desorption isotherms and c pore-size distributions of PEI-functionalized MIL−101(Cr) synthesized under different impregnation times (conducted with filtration, see Methods). d High-angle annular dark-field scanning transmission electron microscope images of ultrathin cuts from PEI-functionalized MIL−101(Cr) at different impregnation times and their corresponding N and Cr elemental mappings.

significant portion of its pore volume. Subsequently, the amine loading rate decelerated, requiring approximately 4 h to fill the remaining pore volume of 0.63 cm³/g, a rate 85 times slower than that during the first 5 min. Overall, PEI-1200 penetrated the internal crystals (cages) of MIL−101(Cr) through the hexagonal windows, completing the entire synthesis within 4 h.

To confirm the successful impregnation of PEI-1200 into the MIL−101(Cr) cages, we investigated the relationship between loaded PEI and MIL−101(Cr). Figure 1d displays a series of high-magnification transmission electron microscopy (TEM) images, scanning TEM (STEM) images, and energy dispersive X-ray spectroscopy (EDX) maps of 50-nm-thick cuts of PEI-functionalized MIL−101(Cr) at specific time intervals. In each instance, elemental Cr and N corresponded to the support and amines. All sample cuts exhibited uniform distributions of Cr and N, with notable variations in the elemental ratios. During impregnation, the N-to-Cr element ratio reached 49:51 within the first 5 min, after which the rate of increase slowed, with a slight shift to 51:49 after an additional 5 min. This observation aligned with the porosity results and confirmed that PEI penetrated the crystal interior, indicating rapid diffusion followed by slow filling stages. To understand the amine distribution within the crystal throughout the loading process, EDX line scans were collected for all the slices (Supplementary Fig. 5). The N content remained nearly constant at different positions within the samples at each stage, indicating a uniform distribution of PEI-functionalized MIL−101(Cr) and the absence of aggregation. In contrast to the approach of impregnating amine-functionalized adsorbents via mesoporous supports[27,30], we demonstrated amine impregnation into the microporous support driven by the abundance of acid sites in the internal crystals. Additionally, we illustrated the uniform dispersion of polymeric amine within MIL−101(Cr) cages, establishing IACI.

## CO₂ capture properties

The uniformity of amine dispersion substantially influences CO₂ uptake, adsorption rates, and regeneration energy for amine-

functionalized adsorbents[31]. We conducted experiments under varied conditions to demonstrate the CO₂ capture properties of PEI-functionalized MIL−101(Cr). The CO₂ mass isothermal adsorption curves (Fig. 2a) illustrate that PEI-functionalized MIL−101(Cr) exhibited rapid adsorption, reaching equilibrium within 15 min at different temperatures. Moreover, the CO₂ uptake decreased as the temperature increased (from 3.2 mmol/g at 30 °C to 1.4 mmol/g at 90 °C), indicating no significant diffusion resistance and thermodynamically controlled CO₂ adsorption. The CO₂ volumetric isothermal adsorption curves (Fig. 2b) align with the isothermal mass adsorption curves, with CO₂ uptake reaching 4.0 mmol/g at 5 °C, further highlighting the CO₂ adsorption performance of PEI-functionalized MIL−101(Cr). Differential scanning calorimetry indicated a regeneration energy consumption of approximately 39.6 kJ/mol of CO₂ (Fig. 2c), substantially lower than other amine-functionalized adsorbents[15,32,33]. Consequently, energy consumption associated with the adsorption-desorption cycles for CCUS would be substantially reduced. In summary, PEI-functionalized MIL−101(Cr) exhibited rapid adsorption equilibrium at room temperature and low energy consumption for desorption. The exceptional CO₂ capture properties observed in PEI-functionalized MIL−101(Cr) could be attributed to the optimized structural changes, which preserve the necessary pathways for CO₂ transportation within the cages containing dispersed PEI molecules. As CO₂ approaches or moves away from the amines, these pathways eliminate potential diffusion barriers, facilitating the adsorption-desorption cycles for CCUS.

Another crucial consideration is amine deactivation during the cyclic process. We conducted 90 cycles of CO₂ adsorption-desorption using a thermogravimetric analyzer regenerated under pure Ar or CO₂ (Fig. 2d). The PEI-functionalized MIL−101(Cr) exhibited high stability and minimal reduction in adsorption capacity during cycles (the negligible decrease is expected due to the slight volatilization of PEI-1200 when exposed to high temperatures)[34]. The decay rates were merely 0.11% and 0.18% per cycle for Ar and CO₂, respectively. This notably outperforms other amine-loaded adsorbents deactivated after a few

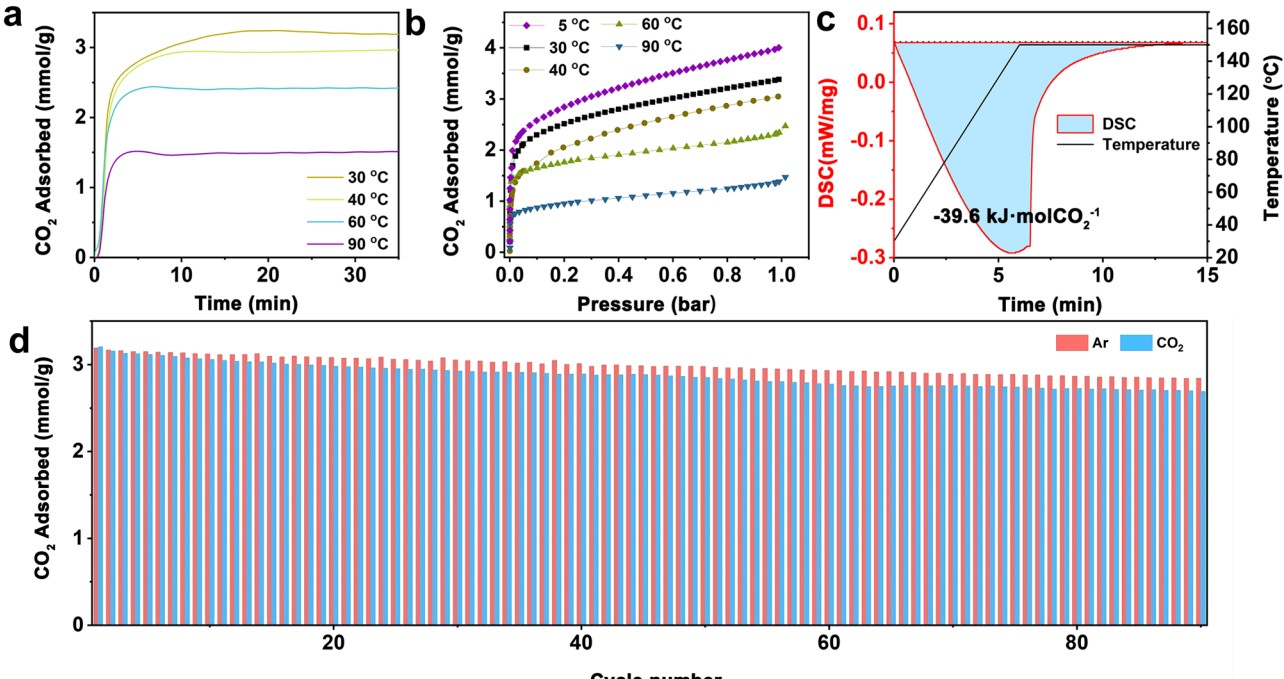

**Fig. 2 | CO₂ capture properties of PEI-functionalized MIL−101(Cr). a** CO₂ isothermal mass adsorption curves at various temperatures (30−90 °C). **b** CO₂ volumetric isothermal adsorption curves at various temperatures (5−90 °C). **c** Differential scanning calorimetry curve of PEI-functionalized MIL−101(Cr)

regeneration. (the blue-shaded region represents the regeneration energy consumption) **d** Extended CO₂ uptakes of PEI-functionalized MIL−101(Cr) with 55 wt.% PEI. (Adsorption in pure 100% CO₂ at 30 °C for 15 min; regeneration in pure 100% Ar or CO₂ at 150 °C for 15 min).

**Table 1 | Comparison of CO₂ capture properties between PEI-functionalized MIL–101(Cr) and previously reported amine-functionalized adsorbents**

| Adsorbents synthesis | | Amine type and content | Ability to sustainable CCUS | | | | | | | Consumption comparison | | Cycle time /min | ref. |
| --- | --- | --- | --- | --- | --- | --- | --- | --- | --- | --- | --- | --- | --- |
| Amine loading | Support | | $X^a$ (mmol/g) | $T_a^b$ /°C | $t_a^b$ /min | $T_d^c$ /°C | $t_d^c$ /min | Purge gas$^d$ | $\eta^e$/% | Amine consumption$^f$ (g/mol$_{CO2}$) | Energy consumption$^g$ (kJ/mol$_{CO2}$) | | |
| impregnated | mesoporous (SiO₂) | PEI-25000 (40 wt.%) | 3.0 | 90 | 30 | 150 | 30 | CO₂ | 8.13 | 20.00 | ~80.22 | 60 | 52 |
| impregnated | mesoporous (carbons) | PEHA$^h$ (78 wt.%) | 3.03 | 75 | 120 | 110 | 120 | N₂ | 1.6 | 8.23 | \ | 240 | 53 |
| impregnated | mesoporous (Al₂O₃) | PEI-1200 (55 wt.%) | 3.0 | 90 | 30 | 165 | 15 | CO₂ | 0.33 | 2.71 | ~92.6 | 45 | 10 |
| impregnated | microporous (MIL–101(Cr)) | PEI-800 (45 wt.%) | 1.1 | 25 | 360 | 110 | 180 | He | 0.9 | 7.28 | 70 | 540 | 14 |
| grafted | mesoporous (SiO₂) | ethane-1,2-diamine (33%) | 1.37 | 75 | 60 | 110 | 10 | N₂ | 0.24 | 1.15 | \ | 70 | 54 |
| grafted | mesoporous (geopolymer) | APTES$^h$ (77 wt.%) | 1.17 | 60 | 60 | 110 | 60 | N₂ | 0.50 | 6.52 | \ | 120 | 8 |
| grafted | mesoporous (zeolite) | EDA$^h$ (16 wt.%) | 1.4 | 40 | 30 | 130 | 30 | CO₂ | 1.05 | 2.37 | \ | 60 | 55 |
| grafted | microporous (Mg₂(dobpdc)) | een$^h$ (36 wt.%) | 3.1 | 80 | 20 | 140 | 10 | CO₂ | 0.30 | 0.70 | ~74 | 30 | 9 |
| IACI | microporous (MIL–101(Cr)) | PEI-1200 (55 wt.%) | 3.2 | 30 | 15 | 150 | 15 | CO₂ | 0.18 | 0.56 | 39.6 | 30 | This study |
| | | | | | | | | Ar | 0.11 | 0.38 | | | |

$^a$X refers to the maximum CO₂ uptake; $^b$$T_a$ and $t_a$ represent the adsorption temperature and time, respectively; $^c$$T_d$ and $t_d$ represent the desorption temperature and time, respectively; $^d$The regeneration atmosphere; $^e$The average inactivation efficiency per cycle; $^f$The amount of amine reagent consumed per mole of CO₂ captured when the CO₂ capacity drops by half; $^g$The regeneration energy consumption at an adsorption and desorption cycle; $^h$PEHA, APTES, EDA, and een represent pentaethylenehexamine, (3-Aminopropyl) triethoxysilane, ethylenediamine, and N-ethylethylenediamine, respectively. (Further details on consumption comparison are provided in Supplementary Note 2).

cycles under CO₂ regeneration because of urea formation[35–37]. It will allow PEI-functionalized MIL–101(Cr) to continuously desorb high-purity CO₂ under CO₂ regeneration atmospheres for subsequent utilization or storage. In comparisons with widely known baseline materials in the field (Supplementary Table 1), the PEI-functionalized MIL–101(Cr) showed obvious advantages (including high capacity, water resistance, high stability, and fast diffusion kinetics).

The high cyclic stability of PEI-functionalized MIL–101(Cr) is likely attributed to Lewis acid sites in MIL–101(Cr), akin to those found in Al₂O₃[12,16]. Fourier-transform infrared (FTIR) spectra of MIL–101(Cr) and PEI-functionalized MIL–101(Cr) supported this notion (Supplementary Fig. 6), revealing a notable decrease in Cr site concentration with amine loading. Given the amine affinity of the support surface, a small fraction may have remained outside the MIL–101(Cr) crystals, experiencing less influence from Lewis acid sites. Consequently, this could lead to lower cycle stability and potential urea formation. The faster decay rate under CO₂ regeneration than under an Ar atmosphere indicates minimal chemical deactivation via urea formation. This could explain the slightly higher decay rate (0.18% per cycle) observed under the former. Table 1 provides comparisons of its CO₂ capture properties with other amine-functionalized adsorbents, highlighting the advantages in three aspects: i) reduced amine consumption for capturing the same amount of CO₂; ii) low regeneration energy consumption; and iii) prolonged stable cycles, thereby extending service life at a reduced cost.

## Difference between IACI and amine impregnation outside crystals

To validate the proposed hypothesis, a series of experiments were conducted to provide robust evidence for the stability of IACI. Initially, amines on the MIL–101(Cr) surface were removed with deionized water. The washed samples exhibited a slight increase in $S_{BET}$ from 7 to 14 m²/g compared to the unwashed samples, as depicted in Fig. 3a. Pore-size distributions indicated minimal changes in 1–2 nm pores after washing (Fig. 3b), indicating that the majority of amines remained bound to the internal supports, with only a negligible fraction being removed during the washing process. Additionally, changes in consumed pore volume (from 1.76 to 1.71 cm³/g) and N content (17.2% to 16.9%) confirmed that the post-washed samples retained approximately 98% of the total amine content (Supplementary Fig. 7), with only 2% washed out. For direct observation, TEM and EDX were employed to image surface-adsorbed amines. Before washing, PEI-functionalized MIL–101(Cr) displayed a series of aggregated particles approximately 500 nm in size (Fig. 3c). The high-magnification TEM image revealed a thin film composed solely of N elements, indicating amine adsorption on the surface, particularly at the edge of the particles. In contrast, post-washed, the thin film composed of only N elements at the particle edges disappeared entirely (Fig. 3d). These findings support the notion that in the designed amine-support system, less than 2% of the amines remained outside the crystals, with the remaining 98% bound within the cages as IACI.

$$2RNH_2 + {}^{13}CO_2 \xrightarrow{>135\,°C} (RNH)_2{}^{13}CO + H_2O \qquad (1)$$

Furthermore, an isotope labeling experiment was conducted during urea synthesis to compare the antiurea formation properties of these two types of amines (Eq. 1). The unwashed PEI-functionalized MIL–101(Cr) was exposed to a pure ¹³CO₂ atmosphere for 24 h (Supplementary Fig. 8), and synthesized urea was detected via time-of-flight secondary ion mass spectrometry in a scanning electron microscope. As depicted in Fig. 3e, the aggregate consisted of regular 500-nm octahedral particles resembling MIL–101(Cr). Subsequently, the ¹³C intensity was observed in the aggregate's planar and vertical directions (Fig. 3f and g). Regarding the surface profile, a homogeneous

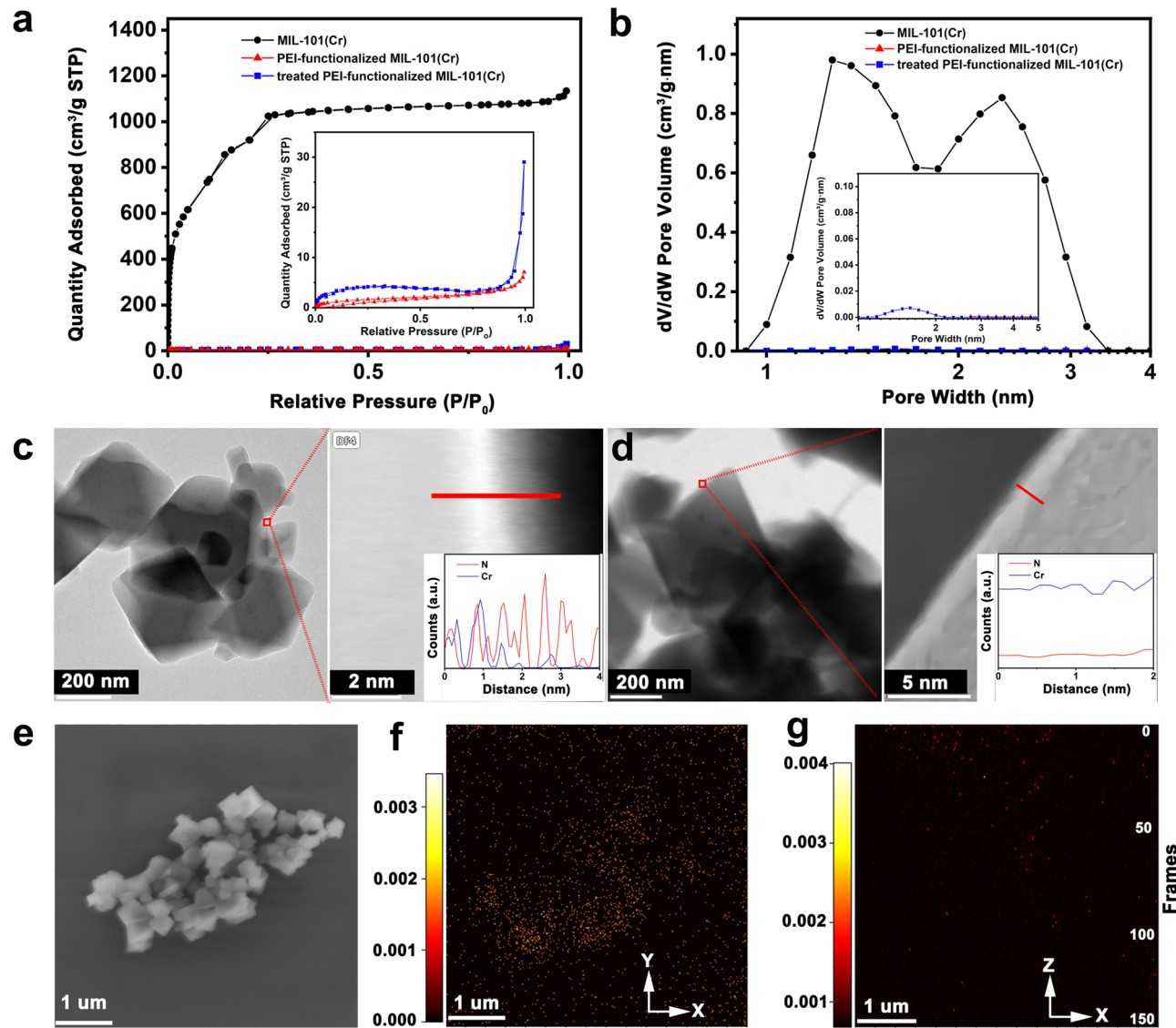

**Fig. 3 | Amine identification and features in PEI−functionalized MIL−101(Cr).**
**a** N$_2$ adsorption/desorption isotherms, and **b** pore-size distributions of the MIL−101(Cr) support for PEI-functionalized MIL−101(Cr) and PEI-functionalized MIL−101(Cr) after washing. High-angle annular dark-field scanning transmission electron microscope images and corresponding energy dispersive X-ray (EDX) spectroscopy maps of N and Cr for **c** PEI-functionalized MIL−101(Cr) and **d** PEI-

functionalized MIL−101(Cr) after washing. (the red lines represent the EDX sites). **e** Cross-sectional scanning electron microscope images of the treated PEI-functionalized MIL−101(Cr) (regeneration under 100% $^{13}CO_2$ at 150 °C for 24 h). Time-of-flight secondary ion mass spectrometry of $^{13}C$ secondary ions (urea distribution) with the spatial topological relation for **f** the interface maps and **g** the depth maps.

distribution of $^{13}C$ was evident on particle outlines, indicating urea formation within the adsorbent and resulting in a small amount of residual adsorbed $^{13}CO_2$ consistent with Fig. 2d. Concerning the depth profile, pronounced aggregation was noted, with the $^{13}C$ signal concentrated on the sample surface, indicating that the urea primarily formed via amines adsorbed on the particle surfaces. To further confirm this conclusion, the relationship between the $^{13}C$ signal intensity and depth was plotted in Supplementary Fig. 9. In the first five frames, the $^{13}C$ signal rapidly decreased from 1.5 to 0.5 and then fluctuated within a specific range. These findings indicate that the cyclic stability of IACI was significantly better than that of amines remaining outside the crystals, indicating a mechanism within IACI that inhibits urea formation.

**Urea inhibition mechanism in IACI**
DFT simulations were conducted to elucidate the urea inhibition mechanism in IACI (see Methods). To balance computational

complexity and accuracy, we simplified the PEI molecule by considering primary, secondary, and tertiary amines (all amine types) as small amine molecules[38]. These molecules then simulated interaction with the Cr site within the super tetrahedron, the fundamental structure of MIL-101(Cr) cages[39]. Coordinatively unsaturated metal sites (CUSs) acted as Lewis acid sites, exhibiting a boiling energy of −1.79 eV (−173 kJ/mol) with the primary amine of PEI (Fig. 4a). This result indicates facile entry and anchoring of amines within the crystal cages of Cr (III) CUSs sites, facilitating high dispersion of IACI. Electron transfer likely occurred upon the interaction of acidic and basic sites. Figure 4b illustrates the differential charge density distribution upon binding primary, secondary, and tertiary amines with Cr sites, indicating substantial electron density redistribution within PEI. Interestingly, electron transfer from amine molecules to the yellow isosurface represents increased charge density, while the light blue isosurface represents decreased charge density). Interestingly, while electron transfer from amine molecules to the MIL−101(Cr) support resulted in an overall positive charge, not all

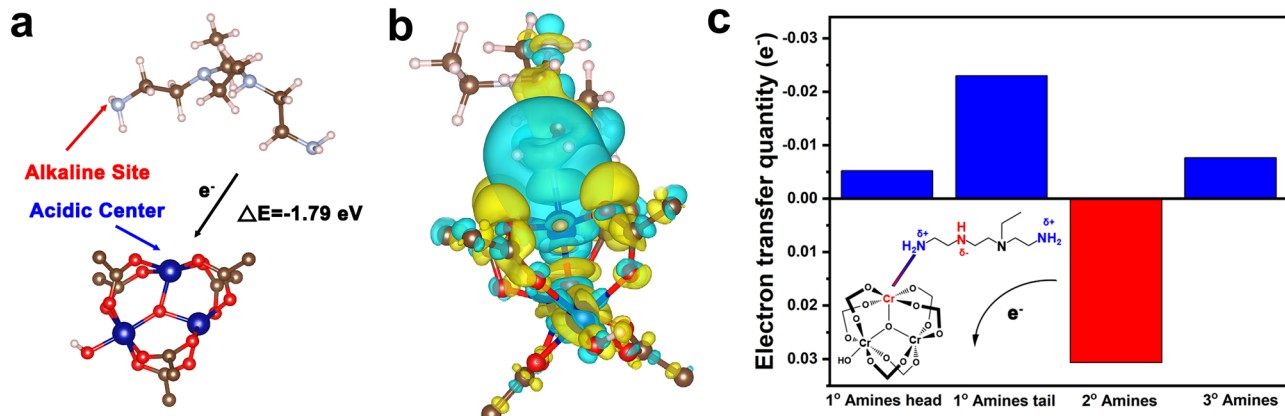

**Fig. 4 | Interaction between PEI and Cr sites. a** Density functional theory calculation for PEI adsorption on the coordinatively unsaturated metal sites of MIL−101(Cr). **b** Differential charge densities of impregnating amines into crystal internalization (yellow indicates increased charge density, light blue indicates decreased charge density). **c** Electron transfer of different types of amines adsorbed on Cr sites. Blue, light gray, brown, red, and white spheres represent Cr, N, C, O, and H, respectively.

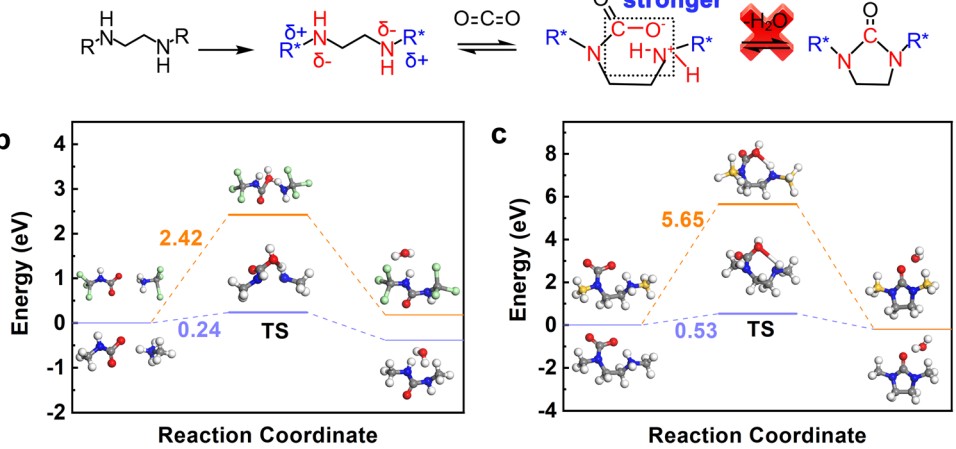

**Fig. 5 | Stability against urea formation. a** Mechanisms of urea inhibition via electronic rearrangement in impregnating amines into crystal internalization. Reaction-energy diagrams of urea inhibition. **b** for mechanism A, and **c** for mechanism B. (Gray, blue, red, white, green, and yellow spheres represent C, N, O, H, Cl, and Si, respectively).

amine groups lost electrons, as revealed by Bader charge calculations. Instead, an electronic rearrangement phenomenon occurred: primary amines lost 0.023 e[-]. In comparison, the secondary amines gained 0.3 e[-] (Fig. 4c). This led to noticeable fluctuations in electron cloud density, warranting solid−state $^{13}C$ nuclear magnetic resonance (NMR) analysis of amine nucleus chemical shifts[40]. Comparison with solid−state $^{13}C$ NMR spectra of 55%PEI@SiO$_2$ revealed decreased proportions of primary and secondary amines in PEI-functionalized MIL−101(Cr) to 27% and 37%, respectively (Supplementary Fig. 10). This evidence provides a stable mechanism resulting from amines·Lewis acid site interaction at the electronic level, a deep insight in the field.

The impact of electron rearrangement on urea inhibition is illustrated schematically in Fig. 5a. When the primary amine loses an electron, this state is simplified to the original R group being replaced by an electron-withdrawing group R′ (δ −). In mechanism A, electron withdrawal results in a partial positive charge (δ +) on the amine nitrogen (N), which attracts electron density from the N − C bond, generating a positive charge (δ +) on the carbon (C) atom. Consequently, interactions between ammonium carbamates are reduced, inhibiting dehydration that leads to urea formation. Mechanism B depicts the effect of electron gains in secondary amines. When the R group is substituted by an electron−donating group R* (δ +), a partial negative charge (δ −) develops on the secondary amine nitrogen. This facilitates electron donation, shifting the electron density of the N − C bond toward the carbon (C) atom, resulting in increased negative charge (δ −). Consequently, the covalent N−H bond is strengthened,

inhibiting breakage during dehydration. Mechanism A was corroborated by reaction-energy diagrams from DMol3 calculations, where the electron-withdrawing group −$CCl_3$ as R' yielded a 2.42 eV transition-state energy, 10 times higher than with the normal R group ($CH_3$), confirming substantial inhibition of urea formation (Fig. 5b). Similarly, mechanism B was validated by considering −$SiH_3$ as R* in Fig. 5c, where the transition-state energy of the intermediate species increased to 5.65 eV, indicating pronounced inhibition. Additionally, an in situ FTIR experiment was conducted to verify the urea formation inhibition under humid conditions (Supplementary Fig. 11). Unlike the urea formation observed in 55% PEI@$SiO_2$, no urea peaks were detected in PEI-functionalized MIL−101(Cr), providing conclusive evidence that IACI does not undergo urea formation even under humid conditions[41].

## Discussion

Amine-functionalized adsorbents exhibit promising $CO_2$ capture performance and are easily prepared, but they encounter challenges such as poor amine dispersion and rapid deactivation, leading to low efficiency and high costs. To overcome these challenges, we have developed an interaction amine-support system characterized by negligible diffusion resistance, high stability, low regeneration energy, and high $CO_2$ adsorption capacity. While some previous amine-impregnated materials have demonstrated excellent properties in specific aspects, they often fall short in other areas[7,42], such as high adsorption capacity but low adsorption rate[20,43], low diffusion resistance but poor stability[23,44], or high stability but moderate $CO_2$ uptake[45,46]. In addition, we have elucidated that the interaction between amines and Lewis acid sites induces a unique electron rearrangement within polymeric amines, resulting in notable urea inhibition in amine-functionalized adsorbents. Regarding the practical implementation, the developed amine-support system offers advantages such as easy scalability, cost-effectiveness in $CO_2$ capture with reduced energy and cost inputs, and feasible industrial applicability due to operational stability. Further research is needed to expand the range of amine-support systems, mainly focusing on small molecule amines and readily available porous supports. This study contributes to the development of an amine-support system to address the challenges of poor amine dispersion and rapid deactivation, with potential for applications in various carbon capture scenarios, including biogas ($CO_2$ and $CH_4$), natural gas ($H_2O$, $O_2$, and $CO_2$), and specific flue gas environments.

## Methods
### Materials
The metal-organic framework MIL−101(Cr) was synthesized following a previously reported procedure[47]. First, 800 mg of Cr $(NO_3)_3$·$9H_2O$ and 332 mg of $H_2BDC$ (terephthalic acid) were blended in 14 mL of deionized (DI) water. After thorough shaking, the mixture was supplemented with 1 mL of 40% HF (hydrofluoric acid). Subsequently, the solution was transferred into a Teflon-lined autoclave and maintained at 220 °C in an oven for 8 h, followed by natural cooling to room temperature. Upon completion of the synthesis, MIL−101(Cr) was isolated via centrifugation and washed repeatedly with MeOH and N, N-Dimethylformamide (DMF) three times each. The MIL−101(Cr) support was obtained by drying under a high vacuum (−0.1 Mpa) at 150 °C for 12 h. In synthesizing the amine-functionalized adsorbent, a physical impregnation method was employed to load the amine into the MIL−101(Cr) support, with an airflow control procedure regulating the impregnation process. First, 0.2 g of MIL−101(Cr) support and 0.24 g of polyethyleneimine were separately placed in two 25 mL beakers and added with 10 mL of methanol. Subsequently, the two mixtures were sonicated at room temperature (25 °C) for 30 min to ensure complete dispersion. Following this, the mixtures were combined and stirred in a fume hood for 4 h with a constant airflow of 1.5 m/s, allowing for the evaporation of methanol. To achieve the desired impregnation time for amine-functionalized MIL−101(Cr), the impregnation reaction

could be terminated before complete methanol evaporation, with the mixture filtered at specific intervals. Finally, the resulting mixture was dried under a vacuum at 60 °C for 12 h to remove methanol completely. For comparison purposes, the same synthetic process for amine-functionalized MIL−101(Cr) was also performed, with MIL−101(Cr) replaced by an equal weight of $SiO_2$ under otherwise identical conditions.

### Characterization
$N_2$ adsorption-desorption analysis of all samples was conducted at −196 °C using a gas adsorption analyzer (Micromeritics, ASAP 2460, USA). Before analysis, MIL−101(Cr) supports, and PEI-functionalized MIL−101(Cr) adsorbents underwent degassing processes at 150 °C for 6 h and 60 °C for 18 h, respectively. The specific surface area ($S_{BET}$) was determined using the Brunauer–Emmett–Teller (BET) equation. At the same time, the total pore volume ($V_{pore}$) and pore-size distribution were assessed using the single-point and DFT models, respectively. Micromorphological observations were conducted using a field emission scanning electron microscope (SEM; Zeiss, Merlin, Germany) and a TEM, TALOSF200, FEI, USA). The TEM instrument was equipped with a high-angle annular dark-field detector and a bright-field detector (1 nA@1 nm, 200 kV). The distribution of $^{13}C$ in PEI-functionalized MIL−101(Cr) adsorbents was analyzed using an ultrahigh-resolution SEM equipped with time-of-flight secondary ion mass spectrometry (TOFSIMS, GAIA3 GMU Model, Czech Republic). Structural characteristics, including phase volume fraction, grain size, and microstructural features, were determined using electron backscattering diffraction analysis. Regeneration energy consumption of the PEI-functionalized MIL−101(Cr) adsorbents was measured using a differential scanning calorimetry analyzer (Discovery DSC, USA) under a ramping temperature of 30–150 °C (20 °C/min) and a $CO_2$ flow of 50 mL/min. Solid-phase $^{13}C$ spectra were acquired using a 600 MHz NMR spectrometer (Agilent, 600 MHz, America) operating at 10 kHz. The $^{13}C$ magic-angle-spinning (MAS) NMR spectra were obtained with acquisition parameters, including an acquisition time of 4 μs, a recycle delay of 5 s, and 300 scans. FTIR spectroscopy was analyzed using a Nicolet iS50 spectrometer (Thermo Fisher Scientific, USA).

### $CO_2$ adsorption tests
The $CO_2$ uptakes of PEI-functionalized MIL−101(Cr) adsorbents were assessed using a thermogravimetric analyzer. First, 15–20 mg of adsorbent was loaded into an alumina crucible and heated from room temperature to 150 °C at a heating rate of 20 °C/min. The sample was then maintained at 150 °C for 30 min for degassing under an Ar flow of 50 mL/min. Following degassing, the temperature was decreased to the desired adsorption temperature of 30–90 °C at a rate of −5 °C/min. Subsequently, the Ar flow was replaced with a $CO_2$ flow of 50 mL/min, and the adsorbing $CO_2$ was carried out for 60 min. Volumetric adsorption isotherms for $CO_2$ were determined using a gas adsorption analyzer (Micromeritics, ASAP 2460, USA) over a temperature range of 5–90 °C. Approximately 150 mg of adsorbent was typically loaded into a glass analysis tube. The sample underwent heating under vacuum at 60 °C for 18 h to remove moisture and adsorbed $CO_2$ from the air. The sample was backfilled with $N_2$ before being transferred to the analysis port. Subsequently, the sample was degassed under vacuum conditions for 4 h, and a high-purity $CO_2$ flow (99.998%) was introduced for $CO_2$ adsorption. For cyclic stability tests, 15–20 mg of adsorbent underwent degassing under the same conditions as the previous degassing process. Upon reaching 30 °C at a rate of −5 °C/min, the Ar atmosphere transitioned to $CO_2$ (50 mL/min), initiating the $CO_2$ adsorption process, which was maintained for 15 min. The temperature was then increased (10 °C/min) to the desired temperature of 150 °C, and a pure $CO_2$ purge (50 mL/min) was introduced to regenerate the adsorbent for 15 min at 150 °C. Subsequently, a new cycle commenced after the temperature was cooled to 30 °C (−5 °C/min).

## Computational details

DFT calculations were performed within the generalized gradient approximation (GGA) using the PBE formulation, employing the Vienna Ab Initio Package (VASP)[48]. Projected augmented wave potentials[49] were utilized to represent the ionic cores and account for valence electrons. In contrast, a plane wave basis set with a kinetic energy cutoff of 400 eV was employed. Partial occupancies of the Kohn-Sham orbitals were considered using the Gaussian smearing method with a width of 0.05 eV, and convergence was achieved when the energy change was less than $10^{-5}$ eV for electronic energy. Geometry optimization was considered converged when the force change was below 0.02 eV/Å. Grimme's DFT-D3 methodology was employed to account for dispersion interactions[50]. The MIL–101(Cr) cluster consisted of 3 Cr, 1 O, and 6 ligands in a cubic box of 22 Å. During structure optimization, the Γ point in the Brillouin zone was used for k-point sampling, allowing all atoms to relax. The adsorption energy ($E_{ads}$) of adsorbate A was calculated using the formula $E_{ads} = E_{A/surf} - E_{surf} - E_{A(g)}$, where $E_{A/surf}$ represents the energy of adsorbate A adsorbed on the surface, $E_{surf}$ is the energy of the clean surface, and $E_{A(g)}$ is the energy of an isolated A molecule enclosed in a cubic periodic box with a side length of 20 Å and a $1 \times 1 \times 1$ Monkhorst-Pack k-point grid for Brillouin zone sampling. This approach maintains alignment with the method's style of nature sustainability while ensuring that it is dissimilar to previous literature. Electrostatic potential computations were performed using the DMol3 code, employing the BLYP exchange-correlation functional within the GGA to describe electron interactions. The double numerical polarization function basis set with an orbital cutoff of 3.6 Å was utilized to expand molecular orbitals, and all core electrons were included in the calculations. Geometry optimizations allowed relaxation of atomic positions, with convergence thresholds set $10^{-5}$ Hartree for energy change, 0.002 Hartree/Å for maximum force, and 0.005 Å for maximum displacement between optimization cycles. The electronic self-consistent field convergence threshold was set to $10^{-6}$ Hartree. The climbing image nudged elastic band method (CI-NEB) was employed to search for minimum energy paths.

## Data availability

The data generated in this study have been deposited in the Figshare database in ref. 51. Source data are also provided in this paper.

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

## Acknowledgements

The authors gratefully acknowledge the generous financial support provided by the National Science Fund for Distinguished Young Scholars (Grant No. 52225407), the National Natural Science Foundation of China (Grant No. 22008104) and Guangdong Provincial Key Laboratory of Soil and Groundwater Pollution Control (No. 2023B1212060002). Additionally, the authors would like to express their gratitude to the Shenzhen Science and Technology Innovation Committee for their financial support through grants JSGG20210713091810035, KCXST20221021111208018 and KCXFZ20211020174805008. These funding sources have played a crucial role in facilitating the research presented in this paper.

## Author contributions

The project was conceived and supervised by Z.Z. The amine-support system was designed by Z.Z. and X.S. Experimental work and data analysis were carried out by X.S. and X-H.S., while density functional theory (DFT) computational simulations were performed by H.W. and F.Y. The manuscript was primarily written by X.S. and Z.Z. Editing and supervision of the research were provided by X.S., G.L, J.H. and Z.Z.

## Competing interests

The authors declare no competing interests.
