## [Peer Review File · Nature Communications]

Atom-level interaction design between amines and support for achieving efficient and stable CO₂ captureREVIEWER COMMENTS

Reviewer #1 (Remarks to the Author):

Overall Review:

The paper investigates the CO₂ capture properties of polyethyleneimine (PEI)-functionalized MIL-101(Cr) adsorbents. It examines the impact of amine dispersion on CO₂ uptake, adsorption rates, and regeneration energy consumption. Through experimentation, the authors demonstrate the rapid adsorption equilibrium, thermodynamically controlled CO₂ adsorption, and low regeneration energy consumption of PEI-functionalized MIL-101(Cr). The cyclic stability of the adsorbent is also evaluated, showing minimal decay rates over multiple cycles. Additionally, the study explores the stability of in-cage amine immobilization (IACI) compared to amine impregnation outside crystals, providing insights into urea formation inhibition mechanisms. However, it could benefit from further discussion on structural changes, exploration of practical applications, and a more comprehensive comparison with baselines to enhance its impact and relevance. While there are minor areas for improvement, the study presents valuable insights and innovations in the field of CO₂ capture.

Questions to authors and suggestions for rebuttal:

1. While the paper provides evidence for the stability of IACI, it lacks discussion on potential structural changes in MIL-101(Cr) due to amine functionalization and their implications for adsorption properties. Addressing this aspect would enhance the mechanistic understanding of CO₂ capture. Could you provide insights into potential structural changes in MIL-101(Cr) due to amine functionalization and their effects on adsorption properties?
2. Although the paper compares the CO₂ capture properties of PEI-functionalized MIL-101(Cr) with other adsorbents, it could benefit from a more comprehensive comparison with widely known baselines in the field, such as commonly used amine-functionalized materials or commercially available carbon capture technologies. Would you consider expanding the comparison of PEI-functionalized MIL-101(Cr) with widely known baselines in the field to provide a more comprehensive assessment of its performance and competitiveness?
3. While the paper discusses the advantages of the developed amine-support system, it lacks in-depth discussion on the practical implementation of the proposed solution, including considerations related to scalability, cost-effectiveness, and industrial applicability. Providing insights into these aspects would enhance the applicability of the research findings. Could you discuss potential considerations and challenges related to the practical implementation of the developed amine-support system in industrial-scale CO₂ capture applications?
4. The paper mentions simplification in the DFT simulations to reduce computational complexity, potentially leading to oversimplification of the urea inhibition mechanism. Further clarification on the limitations of the modeling approach and its implications on the validity of the results would be beneficial. How do you ensure the reliability and accuracy of the DFT simulations despite the simplifications made to reduce computational complexity?
5. While the study highlights the innovative nature of the developed amine-support system, it briefly mentions potential future research directions without elaborating on specific areas or challenges that could be addressed. Expanding on these aspects would provide clarity on the next steps in advancing the field of CO₂ capture. Can you elaborate on specific areas or challenges that could be addressed in future research to further enhance the efficiency and applicability of amine-functionalized adsorbents for CO₂ capture?

Reviewer #2 (Remarks to the Author):

This paper demonstrates that PEI impregnated MIL-101(Cr) for CO₂ capture. Additionally, the authors employ a novel approach referred to as amines into crystal internalization (IACI) to

achieve a uniform dispersion of PEI within MIL-101(Cr). A specific amine, PEI-1200, was selected for impregnation due to its optimal molecular size relative to the pore size of the MIL-101(Cr). Additionally, they investigated the CO₂ adsorption capacity of PEI-MIL-101(Cr) at various temperature. The cyclic tests were also performed.

To elucidate the binding sites of the PEI in the MIL-101(Cr), a washing process with H₂O is employed for the PEI-decorated MIL-101(Cr). Following the washing process, it is confirmed that the majority of the PEI is located within the MIL-101(Cr). Furthermore, the authors utilize Density Functional Theory (DFT) calculations to demonstrate that the PEI bonded with the Open Metal Sites (OMS) of the MIL-101(Cr) suppresses the formation of urea.

In my opinion, this paper is rejected owing to a lack of novelty to be published in Nature Communications. Therefore, the authors need to clearly explain the novelty of this work in comparison to previous studies (ACS Sustainable Chem. Eng. 2016, 4, 5761; Scientific Reports, 2013, 3, 1859) and additional information is necessary to account for the observed phenomenon.

1. I don't understand how to obtain the PEI-impregnated MIL-101(Cr) shown in Figure 1. In Figure S4, a new impregnation method is introduced. I am curious whether the PEI-impregnated MIL-101(Cr) obtained at different reaction times are indeed acquired through this new impregnation method. If this is correct, it is necessary to include a detailed explanation of this process in the experimental section.

2. The authors should compare the amine-impregnated MIL-101(Cr) and present the results in Table 2.

3. What are the adsorption conditions in Fig 2d? Additionally, both conditions show a gradual reduction in the adsorption curves, suggesting instability in the sample. The reduced capacity raises suspicion of amine leaching or the formation of urea. Therefore, authors need to provide additional explanation on this point.

4. I don't believe the urea inhibition mechanism though DFT calculation results only. Therefore, the authors should provide experimental data, demonstrating that the sample does not form urea even when exposed to humid conditions. This additional evidence is essential for a comprehensive understanding of the inhibition mechanism.

5. Finally, what distinguishes this paper from other studies?

Reviewer #3 (Remarks to the Author):

The research topic is important to me because it is not only related to CO₂ capture, an important climate change, but also focused on development of a method for increasing CO₂ capture capacity and sorbent stability. The research was well done, from both experimental and thermotical perspectives. The stability of sorbent was demonstrated with 100 cyclic tests. The paper is publishable in NC.

However, the readers need to recheck the grammatic errors in this paper.

e.g., "... uniform dispersions and highly stability" should be "... uniform dispersions and high stability."

RESPONSE TO REVIEWERS' COMMENTS

Reviewer #1

General comment: The paper investigates the CO₂ capture properties of polyethyleneimine (PEI)-functionalized MIL-101(Cr) adsorbents. It examines the impact of amine dispersion on CO₂ uptake, adsorption rates, and regeneration energy consumption. Through experimentation, the authors demonstrate the rapid adsorption equilibrium, thermodynamically controlled CO₂ adsorption, and low regeneration energy consumption of PEI-functionalized MIL-101(Cr). The cyclic stability of the adsorbent is also evaluated, showing minimal decay rates over multiple cycles. Additionally, the study explores the stability of in-cage amine immobilization (IACI) compared to amine impregnation outside crystals, providing insights into urea formation inhibition mechanisms. However, it could benefit from further discussion on structural changes, exploration of practical applications, and a more comprehensive comparison with baselines to enhance its impact and relevance. While there are minor areas for improvement, the study presents valuable insights and innovations in the field of CO₂ capture.

Response: Thank you for your positive comments and valuable suggestions. Accordingly, we have made several modifications to the manuscript. In addition to comparing our materials with typical adsorbents in the field, we have included further discussions on the structural changes and practical applications within the manuscript. We have considered all the comments in depth and revised the manuscript following your advice.

Specific comments:

Comment 1: While the paper provides evidence for the stability of IACI, it lacks discussion on potential structural changes in MIL-101(Cr) due to amine functionalization and their implications for adsorption properties. Addressing this aspect would enhance the mechanistic understanding of CO₂ capture. Could you provide insights into potential structural changes in MIL-101(Cr) due to amine functionalization and their effects on adsorption properties?

Response: Thank you for your valuable comment. The adsorption properties of CO₂ are greatly influenced by the structural changes in the support after amine loading. In earlier studies [*J. Am. Chem. Soc.* 137, 11749-11759 (2015)], there were often significant pore blockage phenomena observed in supports after amine loading, which hindered the entry of CO₂ into the pores and suppressed the adsorption reaction with amines. Therefore, the dispersibility of amines in the support is of utmost importance. In our research, we have successfully demonstrated the penetration of amines through the hexagonal windows and their binding to the Cr sites within MIL-101(Cr). The amines bound to the highly dispersed Cr sites also exhibit noteworthy dispersion characteristics within the crystal cages. The structural changes provide ample pathways for CO₂ transportation, eliminating diffusion resistance and enabling rapid, low-temperature adsorption with low energy consumption during the regeneration process.

Corresponding revisions: Page 5, Lines 154-159, "The exceptional CO₂ capture properties observed in PEI-functionalized MIL-101(Cr) could be attributed to the optimized structural changes, which preserve the necessary pathways for CO₂ transportation within the cages containing dispersed PEI molecules. As CO₂ approaches or moves away from the amines, these pathways eliminate potential diffusion barriers, facilitating the adsorption-desorption cycles for CCUS."

Comment 2: Although the paper compares the CO₂ capture properties of PEI-functionalized MIL-101(Cr) with other adsorbents, it could benefit from a more comprehensive comparison with widely known baselines in the field, such as commonly used amine-functionalized materials or commercially available carbon capture technologies. Would you consider expanding the comparison of PEI-functionalized MIL-101(Cr) with widely known baselines in the field to provide a more comprehensive assessment of its performance and competitiveness?

Response: Thank you for your kind suggestion. To provide a more comprehensive assessment of its performance and competitiveness, we have compared 6 typical baseline materials in the field of CO₂ capture [*Energy Environ. Sci.* 9, 1803-1811 (2016), *J. Am. Chem. Soc.* 137, 4787-4803 (2015), *Chem. Soc. Rev.* 51, 9340-9370 (2022).], including commonly used and commercially available materials. As shown in the Supplementary Table 1, the PEI-functionalized MIL-101(Cr) exhibited obvious advantages (including high capacity, water resistance, high stability, and fast diffusion kinetics).

Corresponding revisions: Page 6, Lines 169-172, "In comparisons with widely known baseline materials in the field (Supplementary Table 1), the PEI-functionalized MIL-101(Cr) showed obvious advantages (including high capacity, water resistance, high stability, and fast diffusion kinetics)."

Supplementary Table 1 | Comparison of CO₂ capture properties between PEI-functionalized MIL-101(Cr) and previously reported CO₂ capture materials. Some detailed characteristics can be found in the ref. 6-8.

Adsorbents		CO ₂ Capture		Characteristics	
Type	Materials	Adsorption temperature (°C)	CO ₂ uptake (mmol/g)	Advantages	Disadvantages ref.
activated carbon (AC)	KOH-treated AC	30	0.84	water resistance; high stability; low recovery	low capacity and selective 9

				energy		
zeolite	NaX	40	2.10	high stability; tunable structure	water sensitive; low capacity under low pressure	10
MOFs	Mg ₂ (dobdc)	40	4.95	high capacity; highly tunable structure	water sensitive; low stability	11
alkali based	CaO	650	9.32	technological maturity; high capacity; water resistance	high recovery energy and temperature input; oxidative degradation;	12
liquid amine	MEA&DMA2P	30	3.06	high capacity; water resistance technological maturity;	expel toxic volatiles	13
typical solid support amine	PEI@SiO ₂	90	3.0	high capacity; water resistance	oxidative degradation; low diffusion kinetics	14
IACI	PEI-functionalized MIL-101(Cr)	30	3.2	high capacity; water resistance; high stability and fast diffusion kinetics	technological immaturity	This study

6. Ozkan, M., Akhavi, A., Coley, W. C., Shang, R. & Ma, Y. Progress in carbon dioxide capture materials for deep decarbonization. *Chem.* **8**, 141-173 (2022).
7. Fu, D. & Davis, M. E. Carbon dioxide capture with zeotype materials. *Chem. Soc. Rev.* **51**, 9340-9370 (2022).
8. Kolle, J. M., Fayaz, M. & Sayari, A. Understanding the Effect of Water on CO₂ Adsorption. *Chem. Rev.* **121**, 7280-7345 (2021).
9. Liu, L., Jin, S., Park, Y., Park, Y. C. & Lee, C. Sorption Equilibria and Kinetics of CO₂, N₂, and H₂O on KOH-Treated Activated Carbon. *Ind. Eng. Chem. Res.* **57**, 17218-17225 (2018).

10. Kim, C., Cho, H. S., Chang, S., Cho, S. J. & Choi, M. An ethylenediamine-grafted Y zeolite: a highly regenerable carbon dioxide adsorbent via temperature swing adsorption without urea formation. *Energy Environ. Sci.* **9**, 1803-1811 (2016).
11. Mason, J. A. et al. Application of a High-Throughput Analyzer in Evaluating Solid Adsorbents for Post-Combustion Carbon Capture via Multicomponent Adsorption of CO₂, N₂, and H₂O. *J. Am. Chem. Soc.* **137**, 4787-4803 (2015).
12. Han, R., Gao, J., Wei, S., Su, Y. & Qin, Y. Development of highly effective CaO@Al₂O₃ with hierarchical architecture CO₂ sorbents via a scalable limited-space chemical vapor deposition technique. *J. Mater. Chem. A.* **6**, 3462-3470 (2018).
13. Wang, L., Tian, X., Fu, D., Du, X. & Ye, J. Experimental investigation on CO₂ absorption capacity and viscosity for high concentrated 1-dimethylamino-2-propanol-monoethanolamine aqueous blends. *The Journal of Chemical Thermodynamics.* **139**, 105865 (2019).
14. Meng, Y. et al. Comprehensive study of CO₂ capture performance under a wide temperature range using polyethyleneimine-modified adsorbents. *J. CO₂ Util.* **27**, 89-98 (2018).

Comment 3: While the paper discusses the advantages of the developed amine-support system, it lacks in-depth discussion on the practical implementation of the proposed solution, including considerations related to scalability, cost-effectiveness, and industrial applicability. Providing insights into these aspects would enhance the applicability of the research findings. Could you discuss potential considerations and challenges related to the practical implementation of the developed amine-support system in industrial-scale CO₂ capture applications?

Response: Thank you for your valuable comment. In practical implementation, scalability, cost-effectiveness, and industrial applicability are of paramount importance as they directly determine the feasibility of a material for applications. We have discussed in detail the potential considerations and challenges related to

the practical implementation of the developed amine-support system in industrial-scale CO₂ capture applications.

1. Easy scalability: Amines are readily available from commercial sources. And the MIL-101(Cr) can be synthesized solvothermally, a method that has been proven as industrially viable for producing MOF materials [*Science*. 374, 1464-1469 (2021)]. Furthermore, the preparation process of the adsorbent is simple, and with further research, it holds potential for large-scale production.

2. Cost-effective CO₂ capture: In comparison to other conventional materials (Table 1 and Supplementary Table 1), the developed amine-support system showcases low regeneration energy requirements, excellent cyclic stability, high CO₂ adsorption capacity, and rapid kinetics. As a result, it enables efficient carbon capture with reduced energy and cost inputs [*Nat. Clim. Chang.* 7, 243-249 (2017)].

3. Feasible industrial applicability: The amine-functionalized adsorbents have no corrosion or toxic safety concerns. Moreover, they can be easily implemented using existing equipment, ensuring practical industrial applicability.

As for the challenges, the cost of MOFs and large molecular amines are generally high than that of the silica-based supports and small amines. Thus, our future focus would be lowering the cost of the adsorbent through synthetic optimization of MIL-101(Cr), its linker, and the large amines, as well as developing more cost-effective alternatives to the support and amines. We have made the following revisions and discussion.

Corresponding revisions: Page 9, Lines 280-285, "Regarding the practical implementation, the developed amine-support system offers advantages such as easy scalability, cost-effectiveness in CO₂ capture with reduced energy and cost inputs, and feasible industrial applicability due to operational stability. Further research is needed to expand the range of specialized amine-support systems, mainly focusing on small molecule amines and readily available porous supports."

Comment 4: The paper mentions simplification in the DFT simulations to reduce computational complexity, potentially leading to oversimplification of the urea inhibition mechanism. Further clarification on the limitations of the modeling approach and its implications on the validity of the results would be beneficial. How do you ensure the reliability and accuracy of the DFT simulations despite the simplifications made to reduce computational complexity?

Response: Thank you for your valuable comment and concern. A PEI-1200 molecule comprises 15 N, 28 C, and 73 H atoms, making its DFT calculations extremely complex and computationally intensive. Therefore, we adopted a simplification approach to reduce computational complexity. Based on the number of nitrogen atoms bonded to hydrogen atoms, PEI can be categorized into three types of amines: primary, secondary, and tertiary amines, where same type amines exhibit similar chemical properties [*J. Am. Chem. Soc.* 134, 13834-13842 (2012); *Nat. Commun.* 12, 1 1949 (2021)]. Many previous studies have classified amines into three types to investigate their properties [*Environ. Sci. Technol.* 50, 1209-1217 (2016); *J. Am. Chem. Soc.* 141, 16590-16594 (2019)]. In our model, this shorter amine molecule has also been categorized into three types and adopts a structure similar to PEI-1200, just with a reduced length compared to PEI. To further demonstrate the urea inhibition, we conducted an *in-situ* FTIR experiment, comparing the samples with conventional PEI@SiO₂ under the same conditions (Supplementary Fig. 11). The FTIR results indicate that the PEI-functionalized MIL-101(Cr) exhibited excellent urea inhibition, providing evidence for strong binding between the amine and Cr sites and further validating the DFT calculation results. Accordingly, we have made the following revisions and explanation.

Corresponding revisions: Page 7, Lines 224-227, "To balance computational complexity and accuracy, we simplified the PEI molecule by considering primary, secondary, and tertiary amines (all amine types) as small amine molecules⁴². These

molecules then simulated interaction with the Cr site within the super tetrahedron, the fundamental structure of MIL-101(Cr) cages⁴³." Page 9, Lines 264-268, "Additionally, an in situ FTIR experiment was conducted to verify the urea formation inhibition under humid conditions (Supplementary Fig. 11). Unlike the urea formation observed in 55% PEI@SiO₂, no urea peaks were detected in PEI-functionalized MIL-101(Cr), providing conclusive evidence that IACI does not undergo urea formation even under humid conditions⁴⁵."

Supplementary Figure 11 | *In situ* FTIR spectra of a 55%PEI@SiO₂ and b PEI-functionalized MIL-101(Cr) adsorbents as a function of adsorption temperature at humid (~5% H₂O) CO₂ stream with the activated sample as the background. the PEI@SiO₂ adsorbent exhibited a significant peak increase at 1706 cm⁻¹, indicating the formation of urea.⁵ On the other hand, no such substance formation was observed for the PEI-functionalized MIL-101(Cr), suggesting its pronounced resistance to urea under humid conditions.

42. Xia, R. et al. Electrochemical reduction of acetonitrile to ethylamine. *Nat. Commun.* 12, 1949 (2021).

43. Liang, J. et al. Encapsulation of a Porous Organic Cage into the Pores of a Metal-Organic Framework for Enhanced CO₂ Separation. *Angewandte Chemie.* 132, 6124-6129 (2020).

45. Li, K. et al. Research on Urea Linkages Formation of Amine Functional Adsorbents During CO₂ Capture Process: Two Key Factors Analysis, Temperature and Moisture. *The Journal of Physical Chemistry C*. 120, 25892-25902 (2016).

Comment 5: While the study highlights the innovative nature of the developed amine-support system, it briefly mentions potential future research directions without elaborating on specific areas or challenges that could be addressed. Expanding on these aspects would provide clarity on the next steps in advancing the field of CO₂ capture. Can you elaborate on specific areas or challenges that could be addressed in future research to further enhance the efficiency and applicability of amine-functionalized adsorbents for CO₂ capture?

Response: Thank you for your constructive and valuable advice. In the specific field of CO₂ capture, the amine-support system exhibits promise for applications in various areas, such as room temperature biogas (CO₂ and CH₄), natural gas (H₂O, O₂, and CO₂), and certain flue gases, owing to its efficient adsorption at room temperature and remarkable stability. However, CO₂ capture faces a complex environment where factors like temperature, water vapor, and acidic impurities can significantly influence the capture efficiency [*Chem. Rev.* 121, 12681-12745 (2021); *Chem. Soc. Rev.* 51, 9340-9370 (2022)], which is also a current focus of our research efforts. We have made the following revisions and discussion.

Corresponding revisions: Page 9, Lines 285-289, "This study contributes to the development of a specialized amine-support system to address the challenges of poor amine dispersion and rapid deactivation, with potential for applications in various carbon capture scenarios, including biogas (CO₂ and CH₄), natural gas (H₂O, O₂, and CO₂), and specific flue gas environments."

Reviewer #2

General comment: This paper demonstrates that PEI impregnated MIL-101(Cr) for CO₂ capture. Additionally, the authors employ a novel approach referred to as amines into crystal internalization (IACI) to achieve a uniform dispersion of PEI within MIL-101(Cr). A specific amine, PEI-1200, was selected for impregnation due to its optimal molecular size relative to the pore size of the MIL-101(Cr). Additionally, they investigated the CO₂ adsorption capacity of PEI-MIL-101(Cr) at various temperature. The cyclic tests were also performed.

To elucidate the binding sites of the PEI in the MIL-101(Cr), a washing process with H₂O is employed for the PEI-decorated MIL-101(Cr). Following the washing process, it is confirmed that the majority of the PEI is located within the MIL-101(Cr). Furthermore, the authors utilize Density Functional Theory (DFT) calculations to demonstrate that the PEI bonded with the Open Metal Sites (OMS) of the MIL-101(Cr) suppresses the formation of urea.

In my opinion, this paper is rejected owing to a lack of novelty to be published in Nature Communications. Therefore, the authors need to clearly explain the novelty of this work in comparison to previous studies (ACS Sustainable Chem. Eng. 2016, 4, 5761; Scientific Reports, 2013, 3, 1859) and additional information is necessary to account for the observed phenomenon.

Response: Thank you so much for your valuable comments. We sincerely appreciate your efforts in reviewing our manuscript. After carefully re-evaluating the previous works [ACS Sustain. Chem. Eng. 4, 5761-5768 (2016); Sci. Rep. 3, 1859 (2013)], we elaborated on providing a more explicit explanation of the novelty and significance of our study in comparison to these studies.

In the previous work, MIL-101(Cr) was used as a support material for amine loading and demonstrated its CO₂ adsorption capacity under various conditions. These studies have significant implications for the application of MIL-101(Cr) in the field of

carbon capture. However, there is a paucity of research on amine dispersion and rapid deactivation, which are crucial factors for effective CO₂ capture. These issues present a significant challenge for amine-functionalized adsorbents.

Our novelty lies in proposing a new design to introduce a collaborative integration between the amine and support for effectively addressing these challenges. In our research, we have successfully demonstrated the penetration of amines through the hexagonal windows in MIL-101(Cr) and their uniform dispersion within the crystal cages, establishing a novel approach known as impregnation of amines into the crystal internalization (IACI). As depicted in Table 1, the amines within the crystals exhibit exceptional CO₂ capture properties. Furthermore, we have elucidated a stable mechanism originating from the interaction of amines with Lewis acid sites at the electronic level. This design strategy of the integrated amine-support system signifies a pioneering exploration utilizing MIL-101(Cr) and advancing the utilization of amine-functionalized adsorbents.

Detailed discussion has been added to the revised manuscript (please see our response to Comment 5). Our point-by-point responses are detailed below.

Specific comments:

Comment 1: I don't understand how to obtain the PEI-impregnated MIL-101(Cr) shown in Figure 1. In Figure S4, a new impregnation method is introduced. I am curious whether the PEI-impregnated MIL-101(Cr) obtained at different reaction times are indeed acquired through this new impregnation method. If this is correct, it is necessary to include a detailed explanation of this process in the experimental section.

Response: We sincerely appreciate the reviewer's thoughtful comments. We acknowledge that our work lacked a clear description of the experimental procedure concerning the adsorbents' different reaction times. In fact, there is a specific filtration step implemented before the methanol evaporation is completed. At a designated time, such as 5 minutes, we terminate the impregnation reaction and promptly filter the MIL-101(Cr) suspended in the amine methanol solution, followed by vacuum drying to obtain the PEI-functionalized MIL-101(Cr) samples synthesized under different impregnation times, as shown in Figure 1. Based on the analysis of BET and TEM results, it is evident that the amines from different reaction times successfully enter the MIL-101(Cr) cages due to their appropriate molecular size and strong interaction with the Cr sites. We have made the following revisions and supplemented details in the experimental section.

Corresponding revisions: Page 10, Lines 304-305 (in the experimental section), "To achieve the desired impregnation time for amine-functionalized MIL-101(Cr), the impregnation reaction could be terminated before complete methanol evaporation, with the mixture filtered at specific intervals." Page 15, Lines 505-507 (in the results section), "pore-size distributions of PEI-functionalized MIL-101(Cr) synthesized under different impregnation times (conducted with filtration, see Methods)"

Comment 2: The authors should compare the amine-impregnated MIL-101(Cr) and present the results in Table 2.

Response: Thanks for the valuable suggestion. Among the limited reports on amine-impregnated MIL-101(Cr), we chose the study by Darunte et al. (*ACS Sustain. Chem. Eng.* 4, 5761-5768 (2016)) as a representative for comparison, as their amine molecule (PEI-800) closely resembles ours, and they have also reported the cyclic stability of the material under gas desorption conditions. As shown in Table 1, even when the desorption conditions were milder at 110 °C, their materials exhibited significantly higher decay rates, amine consumption, energy consumption, and cycle time. This could be attributed to the fact that although they loaded the amine onto MIL-101(Cr), they did not induce the amine to enter the MIL-101(Cr) cages and interact with the Cr sites and therefore uniform dispersion of amine was not achieved.

Corresponding revisions:

Table 1 | Comparison of CO₂ capture properties between PEI-functionalized MIL-101(Cr) and previously reported amine-functionalized adsorbents

Adsorbents synthesis		Ability to sustainable CCUS					Consumption comparison				
Amine loading	Support	Amine type and content	X ^a (mmol/g)	T _a ^b /min	T _d ^c /T _d ^c / °C	Purges ^e /min	η ^f %	Amine consumption (g/mol _{CO2})	Energy consumption ^g (kJ/mol _{CO2})	Cycle time /min	ref.
impregnated	mesoporous (SiO ₂)	PEI-25000 (40 wt.%)	3.0	90	30 150 30	CO ₂	8.13	20.00	~80.22	60	³⁷
impregnated	mesoporous (carbons)	PEHA ^h (78 wt.%)	3.03	75	120 110 120	N ₂	1.6	8.23	\	240	³⁸
impregnated	mesoporous (Al ₂ O ₃)	PEI-1200 (55 wt.%)	3.0	90	30 165 15	CO ₂	0.33	2.71	~92.6	45	¹⁰

IACI	impregnated	microporous PEI-800 (MIL-101(Cr)) (45 wt.%)	1.1	25	360	110	180	He	0.9	7.28	70	540 ¹⁴
	grafted	mesoporous ethane-1,2-diamine (SiO ₂) (33%)	1.37	75	60	110	10	N ₂	0.24	1.15	\	70 ³⁹
	grafted	mesoporous APTES ^h (geopolymer) (77 wt.%)	1.17	60	60	110	60	N ₂	0.50	6.52	\	120 ⁸
	grafted	mesoporous EDA ^h (zeolite) (16 wt.%)	1.4	40	30	130	30	CO ₂	1.05	2.37	\	60 ⁴⁰
	grafted	microporous Een ^h (Mg ₂ (dobpc)) (36 wt.%)	3.1	80	20	140	10	CO ₂	0.30	0.70	~74	30 ⁹
IACI	microporous PEI-1200 (MIL-101(Cr)) (55 wt.%)	3.2	30	15	150	15	CO ₂ Ar	0.18 0.11	0.56 0.38	39.6	30	This study

14. Darunte, L. A., Oetomo, A. D., Walton, K. S., Sholl, D. S. & Jones, C. W. Direct air capture of CO₂ using amine functionalized MIL-101(Cr). *ACS Sustain. Chem. Eng.* 4, 5761-5768 (2016).

Comment 3: What are the adsorption conditions in Fig 2d? Additionally, both conditions show a gradual reduction in the adsorption curves, suggesting instability in the sample. The reduced capacity raises suspicion of amine leaching or the formation of urea. Therefore, authors need to provide additional explanation on this point.

Response: Thank you for your valuable comment. We realized that we overlooked including the adsorption conditions. In Fig. 2d, the 90 cycles were conducted under pure CO₂ adsorption conditions, while the desorption conditions were carried out

separately using pure Ar or pure CO₂. During the desorption with pure Ar, urea cannot be generated, the reduction in the adsorption curves is attributed to the slight volatilization of PEI-1200 [*J. Am. Chem. Soc.* 139, 3627-3630 (2017)]. During the desorption with pure CO₂, not only does the amine tend to volatilize, but it also leads to urea deactivation [*Chem. Soc. Rev.* 48, 3320-3405 (2019)]. Although a significant portion of PEI enters the MIL-101(Cr) cages to suppress urea formation by binding with the Cr sites, a small fraction of surface-bound amines still contributes to the slight decay. Overall, as indicated in Table 1, regardless of the desorption atmosphere, the materials exhibit excellent stability (decay rates of 0.11% and 0.18% per cycle in Ar and CO₂, respectively). We have made the following revisions and explanation.

Corresponding revisions: Page 5, Lines 162-166, "The PEI-functionalized MIL-101(Cr) exhibited high stability and minimal reduction in adsorption capacity during cycles (the negligible decrease is expected due to the slight volatilization of PEI-1200 when exposed to high temperatures)³⁴. The decay rates were merely 0.11% and 0.18% per cycle for Ar and CO₂, respectively." Page 6, Lines 181-182, "This could explain the slightly higher decay rate (0.18% per cycle) observed under the former." Page 16, Lines 514-515, "Adsorption in pure 100% CO₂ at 30 °C for 15 min"

34. Pang, S. H., Lee, L., Sakwa-Novak, M. A., Lively, R. P. & Jones, C. W. Design of Aminopolymer Structure to Enhance Performance and Stability of CO₂ Sorbents: Poly(propylenimine) vs Poly(ethylenimine). *J. Am. Chem. Soc.* 139, 3627-3630 (2017).

Comment 4: I don't believe the urea inhibition mechanism though DFT calculation results only. Therefore, the authors should provide experimental data, demonstrating that the sample does not form urea even when exposed to humid conditions. This additional evidence is essential for a comprehensive understanding of the inhibition mechanism.

Response: Thank you for your valuable suggestion. Water vapor is a ubiquitous component, and its interactions with amine-functionalized adsorbents can be quite complex [Chem. Rev. 121, 7280-7345 (2021)]. On one hand, previous studies have indicated [J. Am. Chem. Soc. 132, 6312-6314 (2010)] that the presence of water vapor inhibits urea formation due to its role in suppressing dehydration reactions. On the other hand, H₂O can also interact with certain active sites [J. Mater. Chem. A, 5, 6794-6816 (2017)], potentially influencing CO₂ adsorption. To further demonstrate the urea inhibition, we conducted an *in-situ* FTIR experiment, comparing the samples with conventional PEI@SiO₂ under the same humid conditions (Supplementary Fig. 11). The FTIR results indicate that the PEI-functionalized MIL-101(Cr) exhibited excellent urea inhibition even in the presence of humidity, providing evidence for strong binding between the amine and Cr sites and further validating the DFT calculation results. Accordingly, we have made the following revisions and explanation.

Corresponding revisions: Page 9, Lines 264-268, "Additionally, an *in situ* FTIR experiment was conducted to verify the urea formation inhibition under humid conditions (Supplementary Fig. 11). Unlike the urea formation observed in 55% PEI@SiO₂, no urea peaks were detected in PEI-functionalized MIL-101(Cr), providing conclusive evidence that IACI does not undergo urea formation even under humid conditions⁴⁵."

Supplementary Figure 11 | *In situ* FTIR spectra of a 55%PEI@SiO₂ and b PEI-functionalized MIL-101(Cr) adsorbents as a function of adsorption temperature at humid (~5% H₂O) CO₂ stream with the activated sample as the background. the PEI@SiO₂ adsorbent exhibited a significant peak increase at 1706 cm⁻¹, indicating the formation of urea.⁵ On the other hand, no such substance formation was observed for the PEI-functionalized MIL-101(Cr), suggesting its pronounced resistance to urea under humid conditions.

45. Li, K. et al. Research on Urea Linkages Formation of Amine Functional Adsorbents During CO₂ Capture Process: Two Key Factors Analysis, Temperature and Moisture. *The Journal of Physical Chemistry C*. 120, 25892-25902 (2016).

Comment 5: Finally, what distinguishes this paper from other studies?

Response: Thanks for your valuable comment. To distinguish our paper from other studies, we have emphasized our design concept and enhanced the precision of our current work.

Previously, researchers [*ACS Sustain. Chem. Eng.* 4, 5761-5768 (2016); *Sci. Rep.* 3, 1859 (2013)] synthesized amine-impregnated MIL-101(Cr) materials for CO₂ capture and primarily evaluated its CO₂ adsorption performance under various conditions. These studies have demonstrated the exceptional CO₂ adsorption capacity of amine-impregnated MIL-101(Cr) and its suitability for different CO₂ concentrations, highlighting their significant implications. However, when considering the properties of CO₂ capture, it is also crucial to address the important issues of low efficiency and substantial costs associated due to poor amine dispersion and rapid deactivation [*Chem.* 8, 141-173 (2022), *Chem. Soc. Rev.* 48, 3320-3405 (2019)]. These factors directly influence material consumption and the feasibility of application. Comparing our materials with the reported PEI(800)-impregnated MIL-101(Cr) [*ACS Sustain. Chem. Eng.* 4, 5761-5768 (2016)] as shown

in Table 1, it is evident that our materials exhibit significantly higher stability and lower energy consumption during regeneration. Notably, our study advances existing research in the following way:

1. Integrated cooperation between amine and support. Unlike previous studies that solely focused on either amine or support, our work aims to develop a specialized amine-support system that carefully matched the molecular structures and chemical properties of the amine and support.

2. Novel approach of IACI. We have introduced IACI for CO₂ capture, where amine molecules directly enter the crystal structure and interact with internal sites, ensuring their uniform dispersion and increased stability within the supports. This approach effectively reduces the expenses associated with amine-functionalized adsorbents by overcoming poor amine dispersion and rapid deactivation during the cycle.

3. Stable mechanism at the electronic level. To explain the structure and resistance to deactivation of IACI, we present the initial evidence of a stable mechanism guided by the interaction between amines and Lewis acid sites at the electronic level.

4. Outstanding CO₂ capture properties overall. Our study introduces a feasible synthesized adsorbent that exhibits exceptional CO₂ capture properties, including low reagent consumption, minimal energy input, and short operation cycles. These findings contribute to the wider application of amine-functionalized adsorbents.

Addressing the concerns raised by the reviewer, we have made revisions to the paper to better highlight the core concept of our current work. We have made the following revisions and explanation.

Corresponding revisions: Page 3, Lines 80-83, "While previous studies have demonstrated the CO₂ capacity of MIL-101(Cr) as a porous support for amine impregnation^{14,29}, they did not delve into the design of an amine-support system with atomic-level precision to address hindered diffusion and amine deactivation."

Page 4, Lines 101-103, "To integrate cooperation between amine and support, we used various molecular-weight PEIs (Supplementary Note 1) to achieve an optimal match between the dimensions of the pores and amines (Supplementary Fig. 3)."

Page 5, Lines 136-137, "Additionally, we illustrated the uniform dispersion of polymeric amine within MIL-101(Cr) cages, establishing IACI."

Page 8, Lines 244-246, "This evidence provides a stable mechanism resulting from amines-Lewis acid site interaction at the electronic level, a deep insight in the field."

14. Darunte, L. A., Oetomo, A. D., Walton, K. S., Sholl, D. S. & Jones, C. W. Direct air capture of CO₂ using amine functionalized MIL-101(Cr). *ACS Sustain. Chem. Eng.* **4, 5761-5768 (2016).**

29. Lin, Y., Yan, Q., Kong, C. & Chen, L. Polyethyleneimine Incorporated Metal-Organic Frameworks Adsorbent for Highly Selective CO₂ Capture. *Sci. Rep.* **3, 1859 (2013).**

Reviewer #3

General comment: The research topic is important to me because it is not only related to CO₂ capture, an important climate change, but also focused on development of a method for increasing CO₂ capture capacity and sorbent stability. The research was well done, from both experimental and thermotical perspectives. The stability of sorbent was demonstrated with 100 cyclic tests. The paper is publishable in NC.

However, the readers need to recheck the grammatic errors in this paper.

e.g., "... uniform dispersions and highly stability" should be "... uniform dispersions and high stability."

Response: Thank you so much for your positive comments. Accordingly, we carefully revised the manuscript and sought assistance from a professional academic editing service to help ensure the avoidance of grammatical errors in this paper (the certificate has been uploaded as a supplementary file).

REVIEWERS' COMMENTS

Reviewer #1 (Remarks to the Author):

The authors have well addressed the questions, and therefore I strongly recommend it for publication in Nature Communications.

Reviewer #2 (Remarks to the Author):

The revised paper has been much improved and I support its publication as is.

RESPONSE TO REVIEWERS' COMMENTS

Reviewer #1 (Remarks to the Author):

The authors have well addressed the questions, and therefore I strongly recommend it for publication in Nature Communications.

Response: We thank the reviewer for his/her positive comments.

Reviewer #2 (Remarks to the Author):

The revised paper has been much improved and I support its publication as is.

Response: We thank the reviewer for his/her positive comments.